# Adaptive Volumetric-Modulated Arc Radiation Therapy for Head and Neck Cancer: Evaluation of Benefit on Target Coverage and Sparing of Organs at Risk

Ciro Franzese [1,2,*,†], Stefano Tomatis [2], Sofia Paola Bianchi [2], Marco Pelizzoli [2], Maria Ausilia Teriaca [2], Marco Badalamenti [2], Tiziana Comito [2], Elena Clerici [2], Davide Franceschini [2], Pierina Navarria [2], Luciana Di Cristina [1,2], Damiano Dei [1,2], Carmela Galdieri [2], Giacomo Reggiori [2], Pietro Mancosu [2] and Marta Scorsetti [1,2]

1   Department of Biomedical Sciences, Humanitas University, Pieve Emanuele, 20090 Milan, Italy
2   Department of Radiotherapy and Radiosurgery, IRCCS Humanitas Research Hospital, Rozzano, 20089 Milan, Italy
*   Correspondence: ciro.franzese@hunimed.eu; Tel.: +39-0282247454
†   Current address: IRCCS Humanitas Research Hospital, Humanitas University, Via Manzoni 56, Rozzano, 20089 Milan, Italy.

**Abstract:** Background: Radiotherapy is essential in the management of head–neck cancer. During the course of radiotherapy, patients may develop significant anatomical changes. Re-planning with adaptive radiotherapy may ensure adequate dose coverage and sparing of organs at risk. We investigated the consequences of adaptive radiotherapy on head–neck cancer patients treated with volumetric-modulated arc radiation therapy compared to simulated non-adaptive plans: Materials and methods: We included in this retrospective dosimetric analysis 56 patients treated with adaptive radiotherapy. The primary aim of the study was to analyze the dosimetric differences with and without an adaptive approach for targets and organs at risk, particularly the spinal cord, parotid glands, oral cavity and larynx. The original plan (OPLAN) was compared to the adaptive plan (APLAN) and to a simulated non-adaptive dosimetric plan (DPLAN). Results: The non-adaptive DPLAN, when compared to OPLAN, showed an increased dose to all organs at risk. Spinal cord D2 increased from 27.91 (21.06–31.76) Gy to 31.39 (27.66–38.79) Gy ($p = 0.00$). V15, V30 and V45 of the DPLAN vs. the OPLAN increased by 20.6% ($p = 0.00$), 14.78% ($p = 0.00$) and 15.55% ($p = 0.00$) for right parotid; and 16.25% ($p = 0.00$), 18.7% ($p = 0.00$) and 20.19% ($p = 0.00$) for left parotid. A difference of 36.95% was observed in the oral cavity V40 ($p = 0.00$). Dose coverage was significantly reduced for both CTV (97.90% vs. 99.96%; $p = 0.00$) and PTV (94.70% vs. 98.72%; $p = 0.00$). The APLAN compared to the OPLAN had similar values for all organs at risk. Conclusions: The adaptive strategy with re-planning is able to avoid an increase in dose to organs at risk and better target coverage in head–neck cancer patients, with potential benefits in terms of side effects and disease control.

**Keywords:** head–neck cancer; adaptive radiotherapy; VMAT; IGRT; image-guided radiotherapy; radiation therapy; oropharynx

## 1. Introduction

Definitive, as well as post-operative, radiotherapy is an essential treatment in the multidisciplinary management of patients affected by head and neck cancer [1]. Recent advances in radiation therapy technology, including intensity-modulated radiation therapy (IMRT) and image-guided radiation therapy (IGRT), yield precise delivery of radiation beams to the target [2] and better sparing of healthy tissues, such as the salivary glands, larynx, oral mucosa and spinal cord [3,4]. However, during the long course of radiotherapy, patients with head–neck cancer may develop significant anatomical changes [5]. These modifications can be related to multiple factors, such as (1) shrinkage of large tumor and/or

nodal masses; (2) weight loss; and (3) resolution of post-operative changes. Different studies have shown that the radiation dose to the target volume and organs at risk can vary significantly due to spatial and volume variability during treatment [6,7]; thus, the re-planning of treatment with adaptive radiotherapy may ensure adequate doses to the target volumes despite these anatomical changes [8]. Prospective trials are now evaluating the clinical benefit of adaptive radiotherapy. The ongoing RadiomicART trial (NCT05081531) is evaluating the benefit from a pre-defined adaptive strategy with multimodal advanced imaging, including magnetic resonance imaging (MRI) scans and positron-emission tomography (PET) scans. However, so far, the evidence on this topic is still scarce and heterogeneous in the majority of cases that include a small sample size. In our study, we aim to investigate, in a large monocenter sample, the dosimetric consequences of adaptive radiotherapy on patients affected by head and neck cancer treated with the volumetric-modulated arc radiation therapy (VMAT) technique, with or without concomitant systemic therapy.

## 2. Materials and Methods

In this single-center retrospective analysis, we included all the patients affected by head–neck cancer treated in our center with adaptive radiotherapy, with or without concomitant systemic therapy, from 2014 to 2021, for both radical and adjuvant intent. The primary aim of the study was to compare the first original plan with a clinical-delivered adapted plan and a simulated dosimetric plan without an adaptive approach. With this comparison, we analyzed the dosimetric differences in the targets and organs at risk (spinal cord, parotid glands, oral cavity and larynx) occurring with and without an adaptive approach.

All the patients included in this analysis were simulated in the supine position and immobilized with a thermoplastic mask. The simulation CT scan with contrast and 3 mm slice thickness was obtained from the frontal sinus to the carina. For the radical setting, the primary tumor and nodal disease were delineated as high-dose clinical target volume (CTV-HD), while elective nodal regions were delineated as low-dose clinical target volume (CTV-LD). The planning target volumes (PTVs) were generated by the expansion of 3 mm from the CTVs. In the adjuvant setting, CTV-HD included the primary tumor bed and the positive nodal levels, while the CTV-LD included the elective nodal regions. All patients were treated with the VMAT technique in its rapid arc form, with daily evaluation of the patient's position with cone-beam CT (CBCT). Organs at risk were contoured according to the guidelines from Bouwer et al. [9] and Christianen et al. [10]. For the dosimetric analysis, two experienced radiation oncologists revised all the contours before re-planning.

The planning aim was to cover 95% of the PTV volume by at least 95% of the prescribed dose. Dose constraints for the main organs at risk are reported in Table 1.

**Table 1.** Dose constraints of the organs at risk included in the study.

| Organs at Risk | Dose-Volume Histogram Metric | Constraint |
|---|---|---|
| Spinal cord | $D_{2\%}$ | <40–45 Gy |
| Parotid glands | $V_{45Gy}$ $V_{30Gy}$ $V_{15Gy}$ Dmean | <24% <45% <67% <26 Gy |
| Oral cavity | $V_{40Gy}$ | <35% |
| Larynx (whole organ) | $V_{40Gy}$ Dmean | <50% <35 Gy |

Patients who were candidates for radical radiotherapy were treated with a total dose of 66–69.96 Gy on the primary tumor and positive lymph nodes and 54 or 54.45 Gy on the elective neck, delivered in 30 or 33 fractions with simultaneous integrated boost. In the adjuvant setting, patients received 60–66 Gy on the primary tumor bed and positive neck levels (depending on the presence or absence of positive margins and/or extracapsular

extension) and 54 Gy on the elective neck, delivered in 30 fractions with simultaneous integrated boost.

Patients included in our study were treated with a pre-defined re-planning strategy in the case of cT3-4 or cN3 stage disease (36, 64.3%), or when relevant weight loss (8, 14.3%) or a shrinkage of the primary tumor and/or the nodal disease was observed during evaluation of the daily CBCT (12, 21.4%). A new simulation CT scan was then performed, together with a new personalized thermoplastic mask, during the third week of the radiotherapy course, followed by re-contouring of all the organs at risk and the target volumes. The adaptive plan (APLAN) was then produced with the modified set of structures and clinically delivered.

For the dosimetric study purpose, three different scenarios were considered as indicative of the expression of the impact of adaptive radiotherapy on the treated patients as follows:

(1)    first simulation CT and original plan (OPLAN);
(2)    second simulation CT and adapted plan (APLAN);
(3)    second simulation CT and original plan (DPLAN).

The first two scenarios refer, respectively, to the dosimetric setting in the pre-adaptation and post-adaptation of therapy, and they reproduce the clinically delivered dose. The third simulated scenario describes patient's dosimetry without plan adaptation (no plan change) but considering all anatomical changes.

A non-parametric statistic (median and quartiles) was applied for the chosen organs at risk selection to describe the dosimetric behavior of the selected variables. Box plots according to Tukey et al. [11] were used to visually represent distributions in the original and adapted setting.

Considering scenario 1 (OPLAN) as the reference, statistical significance of the differences between both scenarios 2 (APLAN) and 3 (DPLAN), with respect to this reference, was evaluated using the Mann–Whitney test. Probability values of $\leq 0.05$ and $\leq 0.01$ were considered significant or highly significant, respectively. In Figure 1, a visual explanation of the comparison of the three plans in a patient affected by locally advanced squamous cell carcinoma of the oropharynx is shown.

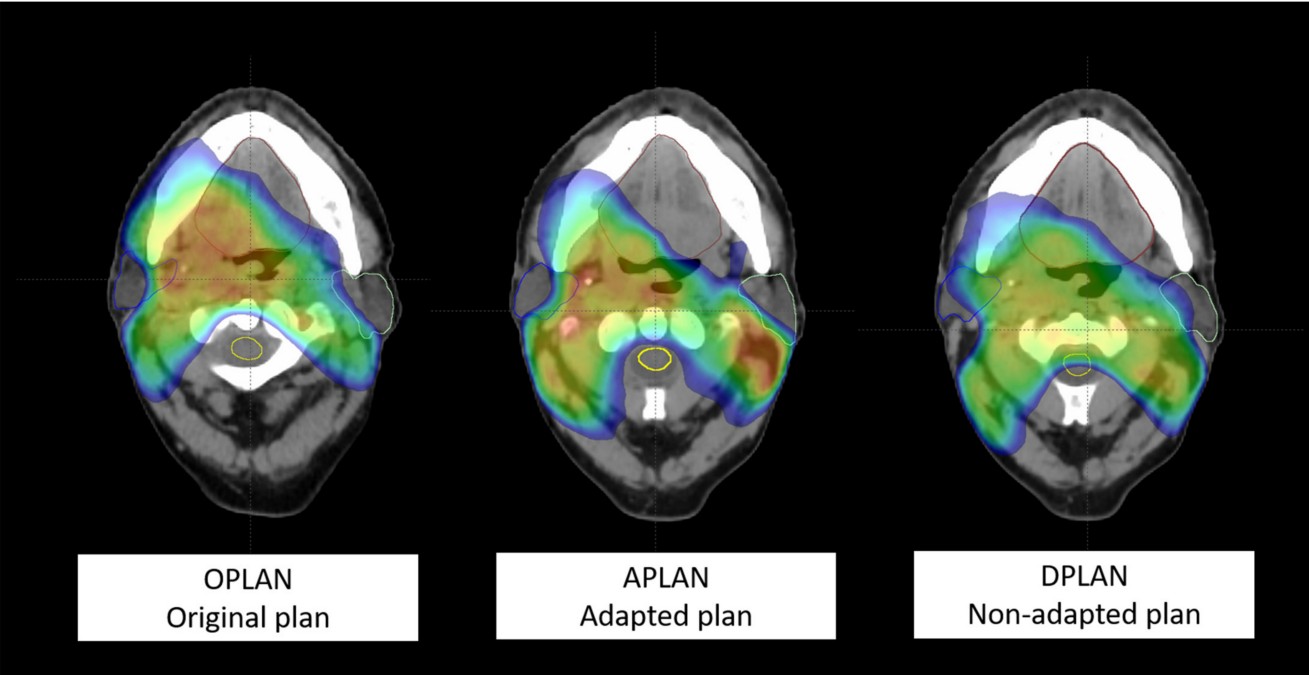

**Figure 1.** Representation of a transverse slice for a patient submitted to adaptive re-planning for a cT3N2c right tonsil squamous cell carcinoma HPV-. From the left to the right: scenario 1 (OPLAN); scenario 2 (APLAN); scenario 3 (DPLAN). The dose ($\geq 35$ Gy) is shown in color wash.

## 3. Results

We included in the present monocenter analysis a total of 56 patients affected by head–neck cancer and treated with adaptive radiotherapy. The characteristics of the included patients and treated diseases are summarized in Table 2. The majority of patients were males (*N* = 36, 64%) and the median age at diagnosis was 69 years (range 39–95). The most common site of primary tumor was the oropharynx (*N* = 17, 30%), followed by the oral cavity (*N* = 14, 25%). Squamous cell carcinoma was the most common histology (*N* = 44, 79%). A total of 21 (37%) patients underwent surgery before radiotherapy, while 35 (62.5%) patients received radical radiotherapy. Thirty-three (59%) patients received cisplatin-based concurrent chemotherapy, while 3 (5.3%) received concurrent. Cetuximab. During radiotherapy and in the 6 months after treatment, patients reported the following acute toxicities: 7 (12.5%) and 1 (1.8%) patients reported grade 1 and grade 2 skin side effects, respectively; 3 (5.3%) patients reported grade 1 mucositis; 8 (14.3%) patients reported grade 1 dysphagia; 12 (21.4%) patients reported grade 1 xerostomia, while 1 (1.8%) patient reported grade 2; and 10 (17.8%) patients reported grade 1 dysgeusia.

**Table 2.** Summary of patient and disease characteristics.

|  | *N. (%)* |
|---|---|
| **Age at Diagnosis, Median (Range)** | **69 years (39–95)** |
| Sex | |
| Male | 36 (64%) |
| Female | 20 (36%) |
| ECOG Performance status | |
| 0 | 15 (27%) |
| 1 | 41 (73%) |
| Smoking | |
| Yes | 22 (39%) |
| Ex-smoker | 5 (9%) |
| No | 29 (52%) |
| Primary tumor site | |
| Oral cavity | 14 (25%) |
| Larynx | 8 (14%) |
| Nasopharynx | 2 (4%) |
| Oropharynx | 17 (30%) |
| Hypopharynx | 5 (9%) |
| Salivary glands | 6 (11%) |
| Thyroid | 2 (4%) |
| Paranasal sinus | 1 (2%) |
| Unknown primary | 1 (2%) |
| Histology | |
| Squamous cell carcinoma | 44 (79%) |
| Other histologies | 12 (21%) |
| Radiotherapy aim | |
| Radical radiotherapy | 35 (62.5%) |
| Adjuvant radiotherapy | 21 (37.5%) |
| Concomitant systemic therapy | |
| Yes | 36 (%) |
| No | 20 (35%) |

Descriptive statistics derived from the evaluation of a dose-volume histogram for the considered organs at risk of our patients are reported in Table 3 for each scenario. Comparisons between the original and adapted context are also shown in the table. Taking scenario 1 (OPLAN) as the reference setting in this analysis, the APLAN had comparable values for all the considered organs at risk. For the spinal cord, $D_{2\%}$ was 27.91 (range

21.06–31.76) Gy for the OPLAN vs. 26.48 (range 22.10–32.51) for the APLAN ($p = 0.66$). Mean dose to the right parotid gland was 20.54 (range 15.86–25.52) Gy and 20.82 (range 17.56–25.67) Gy for the OPLAN and the APLAN ($p = 0.68$), respectively, while mean dose to the left parotid gland was 20.80 (range 15.25–27.24) Gy and 20.68 (range 14.73–26.00) Gy ($p = 0.84$). The mean dose to the larynx (33.47, range 25.33–39.81, for OPLAN vs. 33.9, range 27.24–42.49, for APLAN, $p = 0.56$) and V40 of oral the cavity (33.45, range 08.45–47.70, for OPLAN vs. 37.05, range 14.50–52.65, for APLAN, $p = 0.53$) were also comparable.

**Table 3.** Dose-volume histogram (DVH) for the considered organs at risk.

| Organs at Risk | DVH Metric | S * | Median (P25–P75) | Difference with S1 * | % | $p$ ** |
|---|---|---|---|---|---|---|
| Spinal cord | $D_{2\%}$ | 1 | 27.91 Gy (21.06–31.76) | | | |
| | | 2 | 26.48 Gy (22.10–32.51) | −1.43 Gy | −5.12% | 0.66 |
| | | 3 | 31.39 Gy (27.66–38.79) | 3.48 Gy | 12.46% | 0.00 |
| Parotid right | Dmean | 1 | 20.54 Gy (15.86–25.52) | | | |
| | | 2 | 20.82 Gy (17.56–25.67) | 0.28 Gy | 1.36% | 0.68 |
| | | 3 | 22.14 Gy (17.80–28.20) | 1.6 Gy | 7.78% | 0.30 |
| | $V_{15Gy}$ | 1 | 45.70% (34.70–60.60) | | | |
| | | 2 | 47.50% (36.30–57.15) | 1.8% | 3.93% | 0.74 |
| | | 3 | 66.30% (45.10–78.10) | 20.6% | 45.07% | 0.00 |
| | $V_{30Gy}$ | 1 | 25.50% (14.70–33.90) | | | |
| | | 2 | 26.20% (17.30–34.00) | 1.2% | 4.70% | 0.84 |
| | | 3 | 40.28% (30.30–54.60) | 14.78% | 57.96% | 0.00 |
| | $V_{45Gy}$ | 1 | 12.40% (05.00–18.30) | | | |
| | | 2 | 14.80% (06.70–19.90) | 2.4% | 19.35% | 0.63 |
| | | 3 | 27.95% (20.20–42.40) | 15.55% | 124.40% | 0.00 |
| Parotid left | Dmean | 1 | 20.80 Gy (15.25–27.24) | | | |
| | | 2 | 20.68 Gy (14.73–26.00) | −0.12 Gy% | −0.57% | 0.84 |
| | | 3 | 22.77 Gy (17.22–28.29) | 1.97 Gy% | 9.47% | 0.26 |
| | $V_{15Gy}$ | 1 | 47.00% (30.90–62.40) | | | |
| | | 2 | 45.60% (34.40–58.30) | −1.4% | −2.97% | 0.51 |
| | | 3 | 63.25% (51.60–83.30) | 16.25% | 34.57% | 0.00 |
| | $V_{30Gy}$ | 1 | 26.10% (15.20–37.70) | | | |
| | | 2 | 27.90% (14.90–36.30) | 1.8% | 6.89% | 0.93 |
| | | 3 | 44.80% (34.90–57.70) | 18.7% | 71.64% | 0.00 |
| | $V_{45Gy}$ | 1 | 11.70% (05.70–21.40) | | | |
| | | 2 | 12.60% (05.50–22.90) | 0.9% | 7.69% | 0.88 |
| | | 3 | 31.89% (22.00–41.40) | 20.19% | 172.56% | 0.00 |

**Table 3.** *Cont.*

| Organs at Risk | DVH Metric | S * | Median (P25–P75) | Difference with S1 * | % | *p* ** |
|---|---|---|---|---|---|---|
| Larynx | Dmean | 1 | 33.47 Gy (25.33–39.81) | | | |
| | | 2 | 33.96 Gy (27.24–42.49) | 0.49 Gy | 1.46% | 0.56 |
| | | 3 | 39.00 Gy (32.85–42.50) | 5.53 Gy | 16.04% | 0.02 |
| | $V_{40Gy}$ | 1 | 24.30% (00.00–40.84) | | | |
| | | 2 | 27.60% (00.00–51.50) | 3.3% | 13.58% | 0.41 |
| | | 3 | 72.95% (57.25–98.05) | 48.65% | 200.20% | 0.00 |
| Oral cavity | $V_{40Gy}$ | 1 | 33.45% (08.45–47.70) | | | |
| | | 2 | 37.05% (14.50–52.65) | 3.6% | 10.76% | 0.53 |
| | | 3 | 70.40% (45.10–89.20) | 36.95% | 110.46% | 0.00 |

* S, scenario: 1, OPLAN; 2, APLAN; 3 DPLAN. ** Statistical significance was evaluated considering scenario 1 as the reference using the Mann–Whitney test.

Upon the comparison of non-adaptive scenario 3 with scenario 1, all the organs at risk showed significant differences with an increase in dose for the majority of the studied parameters. For spinal cord, a median D2% of 31.39 (range 27.66–38.79) Gy was observed for the DPLAN vs. 27.91 (range 21.06–31.76) Gy of the OPLAN ($p$ = 0.00). While no variation was observed for the mean dose, a statistically significant difference was observed for V15Gy, V30Gy and V45Gy of the parotid gland. Comparing V15Gy, V30Gy and V45Gy of the DPLAN vs. the OPLAN, an increase of 20.6% ($p$ = 0.00), 14.78% ($p$ = 0.00) and 15.55% ($p$ = 0.00) for the right parotid gland and 16.25% ($p$ = 0.00), 18.7% ($p$ = 0.00) and 20.19% ($p$ = 0.00) for the left parotid gland was observed. Related to the larynx, mean dose was 39.00 (range 32.85–42.50) Gy for the DPLAN vs. 33.47 (range 25.33–39.81) Gy of the OPLAN ($p$ = 0.02), and V40 was 72.95% (range 57.25–98.05) for the DPLAN vs. 24.30 (range 00.00–40.84) of the OPLAN ($p$ = 0.00). Finally, a difference of 36.95% was observed for the V40 of the oral cavity ($p$ = 0.00). Figure 2 shows the plotted dosimetric variable of parotid glans, and Figure 3 shows the plotted dosimetric variable of the spinal cord, larynx and oral cavity. Scenario 2 is excluded from these figures.

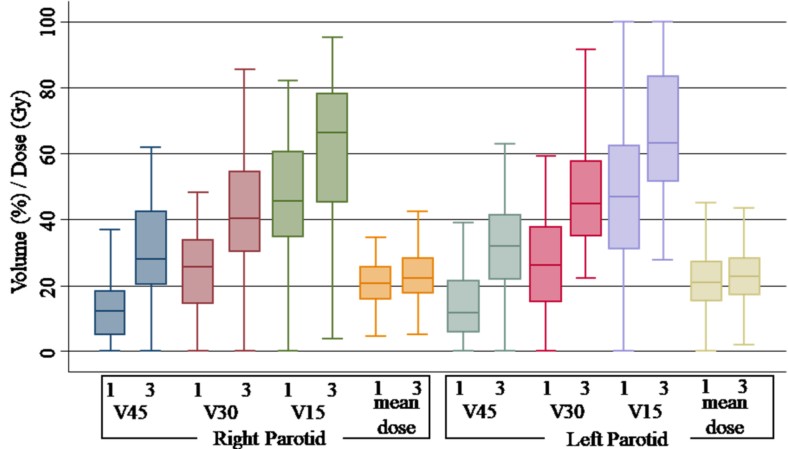

**Figure 2.** Box plot of dose-volume histogram indicators for parotid glands. Different scenarios are indicated as follows: 1, original plan and simulation CT (OPLAN); 3, original plan and second simulation CT (DPLAN). Differences between scenarios were found to be significant ($p$ < 0.01) for all parameters, except mean dose (not significant).

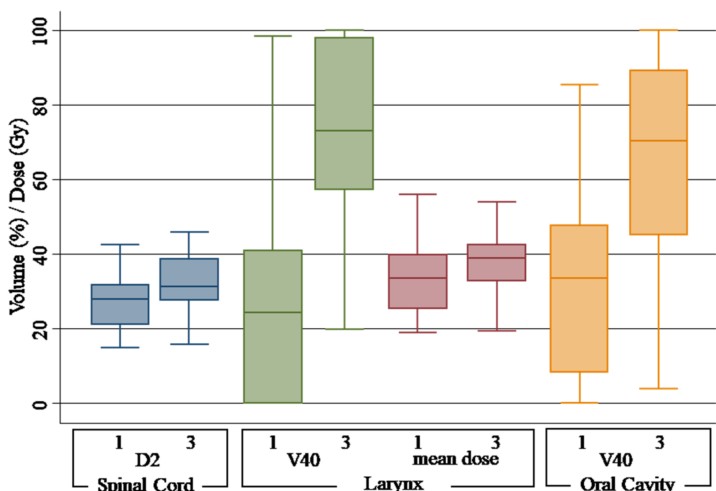

**Figure 3.** Box plot of dose-volume histogram indicators for spinal cord, larynx and oral cavity. Different scenarios are indicated as follows: 1, original plan and simulation CT (OPLAN); 3, original plan and second simulation CT (DPLAN). Differences between scenarios were found to be significant ($p < 0.05$) for all parameters.

The proportion of patients not satisfying our dose constraints is shown in Table 4 for the different organs at risk and set scenarios. No specific differences were observed between scenario 1 (OPLAN) and 2 (APLAN). In contrast, significance of differences was achieved between scenario 1 and 3 (DPLAN), except for mean dose to the parotid glands.

**Table 4.** Percentage of patients with organs at risk constraints out of tolerance according to our protocols (dose-volume histogram = DVH).

| Organs at Risk | DVH Metric | S1 * % | S2 * % | S3 * % | $p$ ** |
|---|---|---|---|---|---|
| Spinal cord | $D_{2\%}$ | 00.00 | 00.00 | 06.25 | 0.060 |
| Parotids, either left or right | Dmean | 42.11 | 44.64 | 53.06 | 0.260 |
| | $V_{45Gy}$ | 31.58 | 35.09 | 87.76 | 0.000 |
| | $V_{30Gy}$ | 21.05 | 15.79 | 69.39 | 0.001 |
| | $V_{15Gy}$ | 24.56 | 21.05 | 55.10 | 0.001 |
| Larynx | $V_{40Gy}$ | 12.96 | 25.45 | 80.56 | 0.000 |
| | Dmean | 43.75 | 46.94 | 69.44 | 0.019 |
| Oral cavity | $V_{40Gy}$ | 46.43 | 51.79 | 77.55 | 0.001 |

* S1, OPLAN; S2, APLAN; S3, DPLAN. ** Statistical significance was evaluated against scenario 1 (OPLAN) using the Chi-squared test. No statistically significant differences occurred between scenario 1 and 2 (APLAN).

With regards to the target coverage, descriptive statistics data are reported in Table 5 for the primary tumor dose. Details of the three considered scenarios are also shown in the table. The comparison of the APLAN with the OPLAN shows the maintenance of both PTV (98.64% vs. 98.72%; $p = 0.35$) and CTV (99.91% vs. 99.96%; $p = 0.30$) coverage. On the contrary, without an adaptive strategy, dose coverage was significantly reduced for CTV (97.90% vs. 99.96%; $p = 0.00$) but, above all, for PTV (94.70% vs. 98.72%; $p = 0.00$).

**Table 5.** Dose-volume histogram (DVH)-derived statistics for target coverage in the 57 head–neck cancer patients in our study.

| Primary Tumor. Target | DVH Metric | S * | Median (P25–P75) % | Difference with S1 * | % | p ** |
|---|---|---|---|---|---|---|
| Planning target volume (PTV) | $V_{95\%}$ | 1 | 98.72 (97.96–99.34) | | | |
| | | 2 | 98.64 (97.25–99.37) | −0.08 | −0.08% | 0.35 |
| | | 3 | 94.70 (87.10–97.60) | −4.02 | −4.07% | 0.00 |
| Clinical target volume (CTV) | $V_{95\%}$ | 1 | 99.96 (99.77–99.99) | | | |
| | | 2 | 99.91 (99.31–99.99) | −0.05 | −0.05 | 0.30 |
| | | 3 | 97.90 (92.33–99.58) | −2.06 | −2.06 | 0.00 |

* S, scenario: 1, OPLAN; 2, APLAN; 3 DPLAN. ** Statistical significance was evaluated considering scenario 1 as the reference using the Mann–Whitney test.

## 4. Discussion

In our retrospective analysis, we demonstrated the relevance of an adaptive strategy for head–neck cancer patients treated with radical or post-operative radiotherapy. Our results show that, in the absence of re-planning, doses to the analyzed organs at risk increase during the long course of radiotherapy delivered with the VMAT technique, with a potential clinical impact in terms of increased toxicity.

A previous study conducted by Hansen et al. [6] produced a similar comparison on thirteen head–neck cancer patients treated with the IMRT technique. The authors demonstrated that without re-planning, doses to 95% (D95) of the PTV of the gross tumor volume and the CTV were reduced in 92% of patients by 0.8 up to 6.3 Gy ($p = 0.02$) and 0.2–7.4 Gy ($p = 0.003$), respectively. Furthermore, the maximum dose (Dmax) to the spinal cord increased in all patients by 0.2 to 15.4 Gy. In another analysis, the study of Schwartz et al. [12] evaluated the benefit of adaptive radiotherapy in 22 patients affected by oropharynx cancer. Four planning scenarios were compared, and the use of IGRT with one instance of adaptive re-planning reduced the mean dose to the contralateral parotid by 0.6 Gy (2.8%, $p = 0.003$) and the mean dose to the ipsilateral parotid by 1.3 Gy (3.9%, $p = 0.002$) over IGRT alone without re-planning.

In our analysis, we observed a significant increase in the dose to the spinal cord, parotid glands, larynx and oral cavity. The most important deviation was observed for the dose received by the larynx, with an increase in V40Gy by 200.20% (from 24.30% to 72.95%) and in mean dose (Dmean) by 16.04% (from 33.47 Gy to 39.00 Gy). The spinal cord had a dosimetric increase of 12.46% in D2 without re-planning.

One of the most studied organs at risk, whose dose seems to be largely affected by adaptive radiotherapy, is represented by the parotid gland [13,14]. Indeed, xerostomia is a common side effect for head–neck cancer patients treated with radiotherapy. The reduction in salivary flow related to radiations [15–17] can negatively impact patient's quality of life (QoL), even for a long time after radiotherapy. Sparing of the parotid glands during head–neck radiotherapy is very challenging due to their close proximity to the main primary tumors, such as the oropharynx, oral cavity, or upper neck levels that require high curative doses. Adoption of advanced radiotherapy techniques, including IMRT and IGRT, should be considered the minimum requirements for better sparing of these organs at risk, above all in cases of small-volume parotids. Throughout the majority of the study, radiotherapy for head–neck cancer demonstrated an increase in the dose received by the parotids, with a variation of up to 6 Gy [18–22]. The study of Castelli et al. [23] demonstrated an average

increase in the parotid gland mean dose of 3.7 Gy. Moreover, the work of Wu et al. [8] showed an increase of about 10% in Dmean. In the study conducted by Jensen et al. [24], the median dose increased in the ipsilateral parotid by 3.87%, and in the contralateral parotid, it increased by 11.5%.

Adaptive radiotherapy seems to be able to compensate the exceeding dose received by the parotid glands during the long course of radiotherapy, particularly in case of tumor shrinkage and neck thickness reduction [23]. Our results demonstrate a clear dosimetric benefit from adaptive radiotherapy for both parotid glands. While the majority of studies evaluate the differences in terms of mean dose, we observed major deviations in volume dose constraints. Without re-planning, the dose constraints for parotids were frequently exceeded, and the deviation was greater the higher the reference dose. The difference for V15Gy, V30Gy and V45Gy was 35–45%, 58–72% and 124–172%, respectively, for the DPLAN, versus 3–3.9%, 4.7–6.9% and 7.7–19.3%, respectively, for the APLAN. Even if not significant, the increase in terms of mean dose was 1.6 Gy and 1.9 Gy for the right and left parotids, respectively. As shown by the analysis of Duma et al. [25], we did not observe a reduction in Dmean of the parotids with adaptive radiotherapy, while a reduction from 0.6 Gy to 4.1 Gy was observed in other experiences [18,23,26–29].

We acknowledge the limitations of the present analysis, which include the retrospective nature and the small number of patients, as well as the heterogeneity of the sample and delivered treatments.

## 5. Conclusions

Within the present study, we evaluated the dosimetric change in organs at risk in patients treated with radiotherapy for head–neck cancer. The adoption of an adaptive strategy with re-planning during the long course of radiotherapy makes it possible to avoid an increase in dose to important organs at risk with potential benefits in terms of side effects and patients' QoL. Prospective randomized trials will be able to better define the benefits of adaptive radiotherapy in head–neck cancer, both in terms of timing and patient selection.

**Author Contributions:** Conceptualization: C.F., M.S.; methodology L.D.C., G.R., P.M.; formal analysis: M.P., S.T.; data curation S.P.B., M.P., D.D., C.G.; writing—original draft: C.F., T.C., E.C., D.F.; writing—review and editing: M.A.T., M.B., P.N.; supervision: M.S. All authors have read and agreed to the published version of the manuscript.

**Funding:** This research received no external funding.

**Institutional Review Board Statement:** The study was conducted in accordance with the Declaration of Helsinki, and approved by the Ethics Committee of Humanitas Research Hospital (protocol code 2986 4 August 2021).

**Informed Consent Statement:** Informed consent was obtained from all subjects involved in the study.

**Data Availability Statement:** The data presented in this study are available on request from the corresponding author.

**Conflicts of Interest:** All the authors declare no conflict of interest.

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
