# Peer review of "Adaptive Volumetric-Modulated Arc Radiation Therapy for Head and Neck Cancer: Evaluation of Benefit on Target Coverage and Sparing of Organs at Risk"

_curroncol, doi:10.3390/curroncol30030254_

Round 1
Reviewer 1 Report
This a current and interesting topic, as the reduction of the side effects rate due to adjuvant therapy should be one of the most important goals for oncologists.
I suppose my only objection would be the division of the parotid glands into “ipsilateral” and “controlateral”, rather than “right” and “left”, supposing that not all the tumors arose in the median line.
Nonetheless, the study is well-conducted and the results are properly supported by the existing literature. In my opinion, the paper would gain even greater value if the authors briefly explain why it is so difficult to spare parotid glands and expose possible solutions.
Author Response
This a current and interesting topic, as the reduction of the side effects rate due to adjuvant therapy should be one of the most important goals for oncologists. I suppose my only objection would be the division of the parotid glands into “ipsilateral” and “controlateral”, rather than “right” and “left”, supposing that not all the tumors arose in the median line.
ANSWER: we agree with the reviewer and indeed the first version of our analysis divided parotid glands in ipsilateral and contralateral. However, we obtained results that were difficult to interpret as most of the patients were irradiated with high doses on the superior neck bilaterally given the advanced loco-regional extension of the disease. For this reason we decided to repeat the analysis dividing the parotids into right and left for a simpler interpretation of the results. We are now conducting a prospective trial on adaptive radiotherapy and planned to divide parotids in ipsilateral and contralateral as suggested. Thank you for the suggestion.
Nonetheless, the study is well-conducted and the results are properly supported by the existing literature. In my opinion, the paper would gain even greater value if the authors briefly explain why it is so difficult to spare parotid glands and expose possible solutions.
ANSWER: We now explained in the discussion why it’s difficult to spare parotids in HN radiotherapy, and possible solutions in terms of minimum requirements and patients’ selection.
Reviewer 2 Report
The authors write an interesting paper on adaptive radiotherapy in patients with head and neck cancer. The paper is well written but requires some minor revisions for spelling and language in general.
Hereunder there are minor considerations that, in my opinion, could improve the quality of the work:
- In the material and methods paragraph the authors would describe the reasons why they performed adaptive radiotherapy in the 56 analyzed patients (% of tumor shrinkage, % weight loss etc.)
- What guidelines were followed to contour the OAR? it would be appropriate to specify if the cases were re-contoured or checked by experienced radiation oncologist before the dosimetry analysis
- For the parotid dosimetry I think is more appropriate the evaluation of “ipsilateral” and “contralateral”
- About oral cavity: explain the evaluation of V40. It is generally used Dmean for oral cavity evaluation
- About Larynx: the authors considered whole larynx? Explicit
- Fig3: Please provide details on the characteristics of the patient represented, such as primary T, staging, etc.
Author Response
The authors write an interesting paper on adaptive radiotherapy in patients with head and neck cancer. The paper is well written but requires some minor revisions for spelling and language in general.
ANSWER: we revised the language of manuscript.
Hereunder there are minor considerations that, in my opinion, could improve the quality of the work:
- In the material and methods paragraph the authors would describe the reasons why they performed adaptive radiotherapy in the 56 analyzed patients (% of tumor shrinkage, % weight loss etc.)
ANSWER: we added this information in the Material and methods paragraph
- What guidelines were followed to contour the OAR? it would be appropriate to specify if the cases were re-contoured or checked by experienced radiation oncologist before the dosimetry analysis
ANSWER: the following references were included in the manuscript for the OARs conturing.
Brouwer CL, Steenbakkers RJHM, Bourhis J, Budach W, Grau C, Grégoire V, et al. CT-based delineation of organs at risk in the head and neck region: DAHANCA, EORTC, GORTEC, HKNPCSG, NCIC CTG, NCRI, NRG Oncology and TROG consensus guidelines. Radiother Oncol 2015;117:83–90. doi:10.1016/j.radonc.2015.07.041.
Christianen MEMC, Langendijk JA, Westerlaan HE, Van De Water TA, Bijl HP. Delineation of organs at risk involved in swallowing for radiotherapy treatment planning. Radiother Oncol 2011;101:394–402. doi:10.1016/j.radonc.2011.05.015.
Moreover, we now specified that the contouring were checked by two experienced radiation oncologists for the dosimetric study.
- For the parotid dosimetry I think is more appropriate the evaluation of “ipsilateral” and “contralateral”
ANSWER: we agree with the reviewer and indeed the first version of our analysis divided parotid glands in ipsilateral and contralateral. However, we obtained results that were difficult to interpret as most of the patients were irradiated with high doses on the superior neck bilaterally given the advanced loco-regional extension of the disease. For this reason we decided to repeat the analysis dividing the parotids into right and left for a simpler interpretation of the results. We are now conducting a prospective trial on adaptive radiotherapy and planned to divide parotids in ipsilateral and contralateral as suggested.
- About oral cavity: explain the evaluation of V40. It is generally used Dmean for oral cavity evaluation
ANSWER: we agree with the reviewer that Dmean is nowadays commonly used for planning of head-neck radiotherapy, even in our Institute. However, considering that included patients were also treated from 2015, we decided for this analysis to evaluate V40 as it was historically adopted in our center during the last years (together with V30). Please find below some references regarding this dose-constraint:
Lu S, et al. (2021) Dosimetric Comparison of Helical Tomotherapy, Volume-Modulated Arc Therapy, and Fixed-Field Intensity-Modulated Radiation Therapy in Locally Advanced Nasopharyngeal Carcinoma. Front. Oncol. 11:764946. doi: 10.3389/fonc.2021.
Li K, Yang L, Xin P, Chen Y, Hu QY, Chen XZ, et al. Impact of dose volume parameters and clinical factors on acute radiation oral mucositis for locally advanced nasopharyngeal carcinoma patients treated with concurrent intensity-modulated radiation therapy and chemoradiotherapy. Oral Oncol 2017;72:32–7.
- About Larynx: the authors considered whole larynx? Explicit
ANSWER: we explicated in Materials and methods that whole larynx was considered.
- Fig3: Please provide details on the characteristics of the patient represented, such as primary T, staging, etc.
ANSWER: we now provided the characteristics of the patient represented in the Figure 3. Thank you for the suggestion.
Reviewer 3 Report
The authors conducted a retrospective study on the dosimetry difference between radiating treatment plan re-adapted to the critical target volume and the simulation of the not-adapted plan for the prosecution of the radiotherapy after the third week since the start. The authors found that the adapted plan irradiated the organs at risk less than the not-adapted plan because the latter didn’t consider the changes in anatomy that occurred in patients in response to the radiotherapy. The study is worthy of consideration for publication but needs efforts to improve its readability. The abstract is misleading and should clearly state that the not-adaptive plan is a simulation. In addition, the authors should list the organs at risk in the methods section of the abstract to improve consistency. The materials and methods section of the manuscript is unclear. The authors should distinguish the clinical procedures performed on the patients from the simulations belonging to the study design. The table listing the organs at risk is not labeled as a table in the manuscript. The authors could place figure 3 in the methods section because such a figure provides a visual explanation of how the study scenarios work. The limitations of the study should be listed in the discussion section. The authors should provide the ethical committee code of the study and approval date.
Before publication, the manuscript needs English editing.
In my opinion, the study should be considered for publication after MINOR REVISIONS.
Author Response
The authors conducted a retrospective study on the dosimetry difference between radiating treatment plan re-adapted to the critical target volume and the simulation of the not-adapted plan for the prosecution of the radiotherapy after the third week since the start. The authors found that the adapted plan irradiated the organs at risk less than the not-adapted plan because the latter didn’t consider the changes in anatomy that occurred in patients in response to the radiotherapy. The study is worthy of consideration for publication but needs efforts to improve its readability.
The abstract is misleading and should clearly state that the not-adaptive plan is a simulation.
ANSWER: we clarified that real adaptive plans were compared to simulated non-adaptive plans. Thank you for the suggestion
In addition, the authors should list the organs at risk in the methods section of the abstract to improve consistency.
ANSWER: we added the organs at risk in the abstract.
The materials and methods section of the manuscript is unclear. The authors should distinguish the clinical procedures performed on the patients from the simulations belonging to the study design.
ANSWER: we thank the reviewer for suggestion. We modified the Material and methods paragraph in order to better explain the design of the study.
The table listing the organs at risk is not labeled as a table in the manuscript.
ANSWER: we now provided the label to the table.
The authors could place figure 3 in the methods section because such a figure provides a visual explanation of how the study scenarios work.
ANSWER: We moved the figure 3 in materials and methods. Thank you for the suggestion.
The limitations of the study should be listed in the discussion section.
ANSWER: We now reported the limitations in the Discussion paragraph.
The authors should provide the ethical committee code of the study and approval date.
ANSWER: we now provided the EC informations.
Before publication, the manuscript needs English editing.
ANSWER: we produced a revision of the whole manuscript and corrected any editing issues.